# Effects of Human Deciduous Dental Pulp-Derived Mesenchymal Stem Cell-Derived Conditioned Medium on the Metabolism of HUVECs, Osteoblasts, and BMSCs

**DOI:** 10.3390/cells11203222

**Published:** 2022-10-14

**Authors:** Ryo Kunimatsu, Tomoka Hiraki, Kodai Rikitake, Kengo Nakajima, Nurul Aisyah Rizky Putranti, Takaharu Abe, Kazuyo Ando, Ayaka Nakatani, Shuzo Sakata, Kotaro Tanimoto

**Affiliations:** Department of Orthodontics and Craniofacial Developmental Biology, Hiroshima University Graduate School of Biomedical and Health Sciences, Hiroshima 734-8553, Japan

**Keywords:** dental pulp, culture, vascular endothelial growth factor A, bone regeneration, stem cells

## Abstract

In this study, we assessed the effects of human deciduous dental pulp-derived mesenchymal stem cell-derived conditioned medium (SHED-CM) on the properties of various cell types. The effects of vascular endothelial growth factor (VEGF) in SHED-CM on the luminal architecture, proliferative ability, and angiogenic potential of human umbilical vein endothelial cells (HUVECs) were determined. We also investigated the effects of SHED-CM on the proliferation of human-bone-marrow mesenchymal stem cells (hBMSCs) and mouse calvarial osteoblastic cells (MC3T3-E1) as well as the expression of *ALP*, *OCN*, and *RUNX2*. The protein levels of ALP were examined using Western blot analysis. VEGF blockade in SHED-CM suppressed the proliferative ability and angiogenic potential of HUVECs, indicating that VEGF in SHED-CM contributes to angiogenesis. The culturing of hBMSCs and MC3T3-E1 cells with SHED-CM accelerated cell growth and enhanced mRNA expression of bone differentiation markers. The addition of SHED-CM enhanced ALP protein expression in hBMSCs and MT3T3-E1 cells compared with that of the 0% FBS group. Furthermore, SHED-CM promoted the metabolism of HUVECs, MC3T3-E1 cells, and hBMSCs. These findings indicate the potential benefits of SHED-CM in bone tissue regeneration.

## 1. Introduction

Regenerative medicine has emerged as a new therapeutic alternative to transplantation that enables the regeneration of dysfunctional tissues using tissue-specific stem cells [1]. Indeed, therapies involving the transplantation of tissue-specific stem cells into damaged areas and cytokine therapies that induce the proliferation and differentiation of stem cells present in tissues have been implemented in preclinical studies and in clinical practice [2]. In particular, undifferentiated mesenchymal stem cells (MSCs) have been increasingly studied in treating refractory diseases, as their use does not require an artificial dedifferentiation step. Furthermore, the use of MSCs reduces the ethical and technical issues associated with using embryonic stem cells/induced pluripotent stem cells during autologous transplantation [3,4,5]. The isolation of MSCs from bone marrow tissue was first reported in 1970 [6]. Subsequently, these cells have been isolated from various tissues, including the adipose, umbilical cord, and placental tissues [7]. Undifferentiated MSCs derived from the bone marrow (BMSCs) are only detected in trace amounts, ranging from 0.001% to 0.01% of the cellular component in the bone marrow. In addition to their broad differentiation potential, BMSCs can be cultured, expanded, and differentiated into various cell types after isolation from the bone marrow. Hence, BMSCs have been widely used in various tissue regeneration therapies [7]. However, the isolation of BMSCs requires an invasive procedure to collect bone marrow fluid, which may increase the burden on patients. Therefore, research in this field has shifted to identifying less invasive MSCs sources without ethical concerns [8]. In this regard, dental pulp stem cells (DPSCs) have attracted attention as a promising cell source in tissue engineering [8]. DPSCs exhibit multiple advantages: (1) they are present in hard tissue, (2) they cause less damage to cells, (3) a large number of cells can be harvested, (4) the harvesting procedure is painless and non-invasive, and (5) there are no ethical concerns associated with their use [8].

Miura et al. [9] successfully isolated and cultured stem cells from the deciduous dental pulp of human exfoliated deciduous teeth (SHEDs). SHEDs show a higher proliferative ability than human BMSCs (hBMSCs) and are efficacious in bone regeneration therapy [10,11,12,13]. Additionally, SHEDs are derived from deciduous teeth, which are usually discarded after tooth extraction and therefore represent a readily available cell source, unlike hBMSCs. In our previous study, we compared the bone regeneration potential of SHEDs, human dental pulp-derived MSCs (hDPSCs), and hBMSCs using cranial bone-deficient immunodeficient mice. Notably, SHEDs and hDPSCs exhibited bone regeneration ability similar to that of hBMSCs [14]. Furthermore, SHEDs exhibited enhanced expression of basic fibroblast growth factor (*bFGF*) and bone morphogenetic protein (*BMP2*) genes and also showed higher proliferative ability in vitro than hBMSCs and hDPSCs [15].

Recently, the paracrine action of cytokines secreted by MSCs has been reported. Indeed, MSCs secrete paracrine factors into the culture supernatant during cell culture, and paracrine efficacy and tissue regeneration by MSCs-conditioned medium (CM) have been reported [16,17,18,19,20,21,22,23]. The MSCs-CM contains insulin-like growth factor-1, transforming growth factor-β1, and vascular endothelial growth factor (VEGF), which have been reported to affect the properties and behavior of regenerating osteocytes [16,20]. Similarly, superior bone regeneration has been reported following the administration of BMSCs-CM in parietal bone defects in rats compared with that obtained using BMSCs implantation alone [23]. Moreover, treatment with BMSCs-CM for paired bony defects of the parietal bone in rats promoted superior bone regeneration compared to that seen with the implantation of BMSCs alone.

We also assessed the effects of SHED-CM on bone regeneration. When SHED-CM was administered and SHED grafting was performed to treat bone defects in the parietal bone of immunocompromised mice, both groups showed a markedly higher regenerated bone volume than the control group [24]. Furthermore, accelerated mature osteogenesis was observed in the SHED-CM group compared with that in the SHED and control groups, accompanied by high expression of VEGF and CD31 and enhanced angiogenesis [24]. There have long been advocates of methods for organizational regeneration that combine the three elements of the “Tissue Engineering Triad”, namely cells, scaffold, and growth factors. Recently, however, some researchers have added oxygen and (blood vessels) that provide nutrition, thus forming the “Tissue Engineering Quad”. With this context in mind, we hypothesize that SHED-CM may affect both angiogenesis and osteogenic bone metabolism.

However, no detailed studies have been conducted to determine the effects of SHED-CM on bone regeneration and angiogenesis in vitro. Hence, this study was designed to verify the proliferative and angiogenic potential of SHED-CM on vascular endothelial cells and to investigate the effects of SHED-CM on the proliferation and bone metabolic potential of osteoblasts and hBMSCs.

## 2. Materials and Methods

### 2.1. Cell Isolation and Culture

Pulp tissue was obtained from deciduous teeth extracted from patients requiring orthodontic treatment. SHEDs were isolated and cultured as previously described [14,15,24]. The multilineage potential of SHEDs used in the present study was determined as previously described [14,15]. According to the International Society for Cellular Therapy, the criteria for defining human MSCs are as follows: (1) cells adhere to the plastic container under standard culture conditions; (2) cells are positive for CD73, CD90, and CD105 and negative for CD11b or CD14, CD19 or CD79α, CD34, CD45, and HLA-DR; and (3) cells have the ability to differentiate into osteoblasts, cartilage cells, and fat cells. We confirmed that the cells isolated from the dental pulp were adherent when cultured in a dish. In our previous study [14], in which cells were isolated from the pulp in an identical manner, the cells were positive for CD29, CD44, CD73, CD105, CD146, and STRO-1 and negative for CD34 and CD271. Furthermore, preliminary flow cytometry analysis confirmed that the cells were positive for CD146, CD44, CD90, CD29, and CD73 and negative for CD34 [24]. Our previous study [14] also demonstrated that the isolated SHEDs have the ability to differentiate into osteoblasts, cartilage cells, and fat cells. In the current study, the cells were isolated and cultured in exactly the same manner. Moreover, SHED cells described by Rikitake et al. [25] were isolated and cultured using the same protocol, and we, therefore, defined the isolated cells as SHEDs.

The study was conducted in compliance with the regulations for epidemiological studies at Hiroshima University Hospital (No. E-20-2) and in accordance with the 1964 Helsinki Declaration and its later amendments or comparable ethical standards. Informed consent was obtained from all participants. Cells at passage 6 were used in the in vivo experiments, whereas those at passages 4–7 were used in the in vitro experiments.

Human umbilical vein endothelial cells (HUVECs; CC-2517; Lonza, Basel, Switzerland) were seeded at a density of 1.6 × 10^3^ cells/well in a 96-well plate (FALCON, Franklin Lakes, NJ, USA) and cultured until they reached 60% confluence in endothelial cell growth basal medium-2 (EBM-2; CC-3156; Lonza) supplemented with reagents in the EGM-2 endothelial SingleQuots kit (Lonza; CC-4176) under an atmosphere containing 5% CO_2_.

hBMSCs (Lonza) were purchased and cultured according to the vendor’s instructions.

### 2.2. Preparation of SHED-CM

SHEDs were cultured, and the medium was changed to serum-free alpha-minimal essential medium (α-MEM) supplemented with 0.5 μL/mL penicillin (Meiji Seika Pharma, Tokyo, Japan), 0.24 μL/mL kanamycin (Meiji Seika Pharma), and 1 μL/mL amphotericin (MP Biomedicals, Strasbourg, France) after reaching 70–80% confluence. After 48 h of incubation, the culture supernatant was centrifuged at 390× *g* for 5 min and at 1580× *g* for 3 min. The supernatant (SHED-CM) was enriched 20-fold using a 10-kDa Millipore filter (Millipore EMD, Burlington, MA, USA) and stored at 4 °C or −80 °C until further use.

### 2.3. Analysis of Cell Proliferation Using the Bromodeoxyuridine Immunoassay

HUVECs were treated with SHED-CM in the presence or absence of 10 μg/mL anti-VEGF-A antibody (MAB293; R&D Systems, Minneapolis, MN, USA) in serum-free EBM-2 medium. Cell proliferation was analyzed using a cell proliferation ELISA BrdU kit (Roche Diagnostics, Basel, Switzerland) according to the manufacturer’s instructions. Absorbance of the samples was measured using a microplate reader (Varioskan LUX; Thermo Fisher Scientific, Inc., Waltham, MA, USA) at 370 nm.

### 2.4. Endothelial Tube Formation Assay

The endothelial tube forming assay kit (Cell Biolabs, San Diego, CA, USA) was used to assess the lumen formation of endothelial cells according to the manufacturer’s instructions. HUVECs in the SHED-CM supplementation group were cultured with SHED-CM added to EBM-2; the SHED-CM plus anti-VEGF-A neutralizing antibody addition group was treated with SHED-CM + anti-VEGF-A neutralizing antibody (10 μg/mL; MAB293, R&D Systems) in EBM-2; the VEGF-A addition group was treated with VEGF-A (10 ng/mL) added to EBM-2; and the cells in the control group were cultured in EBM-2 alone. Extracellular matrix (ECM) gel solution prepared from murine sarcoma was added to a cooled 96-well plate and incubated for 60 min to form a gel. HUVECs (5.0 × 10^4^ cells/well) were seeded on an ECM gel solution and cultured at 37 °C under 5% CO_2_. The cells were observed under a microscope (BZ-X800; Keyence, Osaka, Japan) 16 h after the initiation of culture, and four ranges were randomly selected on the plate to measure branch points and vessel length.

### 2.5. Effects of SHED-CM on the Proliferation of Osteoblasts and hBMSCs

The mouse osteoblastic cell line MC3T3-E1 (Riken BRC; Ibaragi, Japan) and hBMSCs (Lonza; PT-2501) were seeded at a density of 1.6 × 10^3^ cells/well in a 96-well plate, and the cells were cultured until they reached 60% confluence. The fetal bovine serum (FBS) concentration in the medium was gradually reduced from 10% to a serum-free state (0%) 12 h before the SHED-CM treatment.

Cells grown under three conditions, namely SHED-CM, 10% FBS, and 0% FBS, were examined. The incorporation of BrdU into the DNA over 24 h was quantified using an ELISA BrdU kit following the manufacturer’s instructions.

### 2.6. Quantitative Real-Time Polymerase Chain Reaction (PCR)

MC3T3-E1 cells and hBMSCs were seeded at a density of 1 × 10^5^ cells/well in six-well plates (FALCON) and cultured in a medium with 10% FBS. Upon reaching 80% confluence and 12 h before treatment with SHED-CM, the cells were deprived of serum in a gradient fashion by culturing in a medium containing 10%, 1%, 0.1%, and 0% FBS. Subsequently, SHED-CM, 0% FBS, or 10% FBS was added to the culture medium, and the cells were then cultured for 48 h. The mRNA levels of alkaline phosphatase (*ALP*), osteocalcin (*OCN*), Runt-related transcription factor (*RUNX2*), and *VEGF* were determined using quantitative real-time PCR analysis with the QuantiTect SYBR Green PCR master mix (Qiagen, Valencia, CA, USA) and LightCycler^®^ 480 II (Roche Diagnostics). Briefly, the total RNA was extracted from the cells using an RNeasy mini kit (Qiagen) and was quantified using a NanoDrop One/One^c^ spectrophotometer (Thermo Fisher Scientific, Inc., Waltham, MA, USA). RNA purity was also assessed with this instrument, based on the OD 260/OD 280 ratio, and only samples with an A260/A280 ratio of 1.5–2.0 were used for further analysis. Then, 1 μg of purified total RNA was reverse-transcribed to cDNA using a ReverTra Ace first-strand cDNA synthesis kit (Toyobo, Osaka, Japan). Real-time PCR was performed using the Thunderbird SYBR qPCR mix (Toyobo) with specific primer sets (Table 1). The relative expression levels were analyzed using the ∆∆Ct method and normalized to those of beta-actin (*ACTB*).

### 2.7. Western Blot Analysis of ALP Protein Expression

MC3T3-E1 cells and hBMSCs were seeded at a density of 1 × 10^5^ cells/well in six-well plates (FALCON) and cultured in α-MEM containing 10% FBS. When cells reached 80% confluence 12 h prior to SHED-CM treatment, FBS concentration was gradually reduced from 10% to 0%. Subsequently, SHED-CM, 0% FBS, or 10% FBS was added to the culture medium, and the cells were then cultured for 48 h. After 24 h of treatment, the cells were lysed using RIPA lysis buffer (Nacalai Tesque, Kyoto, Japan) supplemented with 1% protease inhibitor cocktail and harvested with a cell scraper. The sample was centrifuged at 15,000 rpm for 20 min at 4 °C, and the supernatant was recovered. The protein concentration was measured using a Bio-Rad protein assay kit (Bio-Rad Laboratories, Hercules, CA, USA), and 40 μg protein was mixed with 10% 2-mercaptoethanol and 0.01% bromophenol blue. The sample was boiled at 100 °C for 3 min. Proteins were then electrophoresed on a 10% SDS-polyacrylamide e-PAGEL gel (ATTO, Tokyo, Japan) and transferred to a PVDF membrane using an iBlot gel transfer system (Thermo Fisher Scientific, Inc., Waltham, MA, USA). After transfer, the membrane was immersed in blocking buffer (Block Ace, DS Pharma Biomedical, Osaka, Japan) and agitated at room temperature for 30 min. The primary antibodies were diluted with immunoreaction enhancer solution 1 (which can obtain signal solution, Toyobo) and incubated with the membrane at room temperature for 30 min. After incubation with the primary antibody, the membranes were washed thrice for 5 min with wash buffer (15 mL PBS and 0.1% Tween-20). IRDye^®^ conjugated secondary antibodies were diluted with immunoreaction enhancer solution 2 (Toyobo) and incubated with the membrane at room temperature with agitation for 1 h. Electrochemical fluorescence detection of the target antigen was performed using the Odyssey^®^ Imaging System (LI-COR Biosciences) for NIR fluorescence detection.

### 2.8. Statistical Analyses

All data are presented as the mean ± standard deviation. Comparisons between groups were performed using the Wilcoxon *t*-test, whereas multiple-group comparisons were performed using the Bonferroni–Dunn method. The level of significance was set at *p* < 0.05 or *p* < 0.01.

## 3. Results

### 3.1. Effects of VEGF-A Blockade on the Proliferation Potential of SHED-CM in HUVECs

HUVECs cultured in the SHED-CM group showed considerably higher proliferative ability than the SHED-CM + anti-VEGF-A antibody-supplemented, VEGF-A, or control group. Additionally, HUVECs cultured in the SHED-CM + anti-VEGF-A antibody-supplemented and VEGF-supplemented groups showed considerably higher proliferative ability than the control group (Figure 1).

### 3.2. Effects of VEGF-A Blockade on the Angiogenic Potential of SHED-CM in HUVECs

Angiogenesis was detected in HUVECs cultured in the SHED-CM-supplemented group. Weaker angiogenesis was observed in the SHED-CM + anti-VEGF-A neutralizing antibody-supplemented group than in the SHED-CM- and VEGF-supplemented groups. In the VEGF-supplemented group, increased angiogenesis and more blood vessels with longer vessels were observed compared with those in the SHED-CM-supplemented, SHED-CM + anti-VEGF-A antibody-supplemented, and control groups. In contrast, the formation of blood vessels was less abundant in the control group than in the other groups (Figure 2A). Vascular bifurcation and vessel length were also measured (Figure 2B,C). In the SHED-CM group, both branch point and blood vessel length were markedly higher than those in the SHED-CM + anti-VEGF-A antibody-supplemented and control groups. In the SHED-CM + anti-VEGF-A antibody-supplemented group, no significant differences were observed. In the VEGF-supplemented group, both branch point and vessel length were markedly higher than those in the SHED-CM + anti-VEGF-A antibody-supplemented and control groups (Figure 2B,C).

### 3.3. Effect of SHED-CM on the Proliferative Ability of hBMSCs and MC3T3-E1 Cells

hBMSCs in the SHED-CM group exhibited a considerably higher proliferative ability than cells in the serum-free (0% FBS) and 10% FBS groups. A considerably higher proliferative ability was observed in the 10% FBS group than in the 0% FBS group (Figure 3A). Similarly, MC3T3-E1 cells in the SHED-CM group exhibited notably higher proliferation than those in the 0% FBS and 10% FBS groups. Additionally, notably higher proliferation was observed in the 10% FBS group than in the 0% FBS group (Figure 3B).

### 3.4. Effects of SHED-CM on hBMSC and MC3T3-E1 Gene Expression

With regard to hBMSCs, 10% FBS- and SHED-CM-supplemented groups showed enhanced gene expression of ALP, Runx2, and OCN compared with that in the 0% FBS group (Figure 4A). ALP, Runx2, and OCN gene expression levels in hBMSCs were not markedly different in the SHED-CM-supplemented group compared with the 10% FBS-supplemented group (Figure 4A). In MC3T3-E1 cells, SHED-CM and 10% FBS-supplemented groups demonstrated considerably enhanced gene expression of ALP, Runx2, and OCN compared with that in the 0% FBS group (Figure 4B). Furthermore, a marked enhancement of ALP, RUNX2, and OCN gene expression in MC3T3-E1 cells was observed in the SHED-CM-supplemented group compared with that in the 10% FBS-supplemented group (Figure 4B).

### 3.5. Effects of SHED-CM on the Protein Expression of ALP in the hBMSCs and MC3T3-E1 Cells

Subsequently, we examined the protein expression of ALP in MC3T3-E1 cells and hBMSCs using Western blot analysis. The protein expression of ALP in the hBMSCs was enhanced upon SHED-CM or 10% FBS supplementation compared with that in the 0% FBS groups (Figure 5A). Furthermore, ALP protein expression in MC3T3-E1 cells also increased in response to SHED-CM or 10% FBS exposure compared with the 0% FBS group (Figure 5B).

## 4. Discussion

BMSCs-CM reportedly promotes periodontal tissue regeneration in rats [26], callus lengthening in mice [16], bone regeneration in rats with skull defects [23], and liver tissue regeneration in rats [27,28]. Adipose-derived stem cells (ASCs) promote bone regeneration in rats with skull defects [29] and liver tissue regeneration in mice [30]. hBMSCs-CM contains tissue-regenerative trophic factors such as osteoprotegerin (OPG), VEGF-C, monocyte chemoattractant protein (MCP)-1, MCP-3, IL-3, IL-6, IL-22, and acellular nerve grafts (ANGs), which are involved in the attraction of MSCs and endothelial/endothelial progenitor cells, differentiation of osteoblasts, and angiogenesis [16]. However, harvesting hBMSCs and ASCs involves surgical invasion; hence, in this study, we focused on the use of the SHED culture supernatant, as it is readily available and can be harvested using a less invasive procedure than that needed for hBMSCs and ASCs.

SHED-CM contains several factors such as OPG, TIMP-1, TIMP-2, ANGs, MCP-1, VEGF-A, and several hepatic regenerative factors that promote anti-apoptotic effects, hepatocyte protection, angiogenesis, macrophage differentiation, and hepatic progenitor cell proliferation and differentiation [31]. In addition, protective effects in murine acute myocardial infarction [32], amelioration of rat ischemic brain injury [33], improvement of cognitive function in a mouse model of Alzheimer’s disease [34], peripheral nerve regeneration in rats [35], improvement of impaired glucose tolerance in streptozotocin-induced diabetic mice [36], anti-inflammatory effects in a rheumatoid arthritis mouse model [37], functional recovery after spinal cord injury [38], liver fibrosis [39], and therapeutic effects in a murine acute lung injury model [40] have been reported. In our previous in vivo study, we found that the SHED-CM and SHED engraftment groups presented considerably more bone regeneration than that seen in the control group [24]. We further elucidated that bone metabolism-related factors (OPG, OPN, BMP-2, and BMP-4), angiogenesis-related factors (M-CSF, MCP-1, ANG, bFGF, HGF, VEGF-C, and VEGF-A), and neurotrophic factors (BDNF, β-NGF, GDNF, and NT-3) were abundant in the culture supernatants [24]. Thus, in this study, we validated the effect of VEGF on angiogenesis using SHED-CM in vitro. Our previous study suggested that the level of VEGF-A present in SHED-CM was 161 pg/mL [24]. Based on this report, the use of 20-fold enriched SHED-CM (extracted by a similar procedure) in the current study suggests that the protein level of VEGF-A in our enriched SHED-CM is approximately 3.2 ng/mL.

We examined the effect of SHED-CM on the proliferative ability of HUVECs and performed lumen-forming assays. Blocking VEGF in SHED-CM suppressed the proliferative and lumen-formation abilities of HUVECs, indicating that VEGF plays a role in the angiogenesis promoted by SHED-CM. The addition of BMSCs-CM to HUVECs reportedly promotes luminogenesis [26]. Furthermore, blocking VEGF in MSC-CM resulted in the inhibition of angiogenesis of HUVECs [26]. Similar results were obtained in this study, even though SHED-CM and BMSCs-CM differed in their tissue origin. Notably, the growth of HUVECs treated with SHED-CM where VEGF was blocked was considerably higher than that of the control group. Enhanced growth of HUVECs by factors such as HGFs and bFGF has been reported [41,42]. As these factors contained in the SHED-CM can stimulate the growth of vascular endothelial cells, it is conceivable that the growth of HUVECs may have been similarly stimulated even after the blockade of VEGF activity.

Next, we assessed the proliferation ability and osteogenesis-related gene expression in hBMSCs and MC3T3-E1 cells. Proliferation was remarkably enhanced in hBMSCs and MC3T3-E1 cells in the SHED-CM-supplemented group compared with that in the control group. Additionally, the expression of *ALP*, *OCN*, and *Runx2* was upregulated in hBMSCs and MC3T3-E1 cells in the SHED-CM-supplemented group compared with that in the 0% FBS group. Furthermore, culturing of the hBMSCs and MC3T3-E1 cells with SHED-CM enhanced not only the mRNA expression of *ALP* but also ALP protein levels. 

Previous studies have shown that BMSCs-CM enhances the proliferation ability of rat-derived MSCs and rat periodontal ligament cells [26]. Furthermore, supplementation of BMSCs-CM considerably enhanced the expression of osteogenesis-related genes (*Alp*, *Ocn*, and *Runx2*) and angiogenesis-related genes (*Vegf*, *Ang1*, and *Ang2*) in rat-derived MSCs [26]. In addition, DPSC-CM induces DPSC-osseous differentiation [43,44]. In this study, similar results were obtained, even though the supernatant from SHED-CM and BMSCs-CM differed. Thus, these findings suggest that SHED-CM promotes the proliferation of hBMSCs and MC3T3-E1 cells and differentiation of osteoblasts.

This in vitro study has some limitations. First, in the present study (and based on previous reports), we performed our in vitro investigations using SHED-CM that was enriched 20-fold. However, the dose–response efficacy for SHED-CM enrichment has not been investigated in depth, and the safety of the enriched CM has not been confirmed. In the future, the safety of concentrated SHED-CM as well as the optimum conditions for bone regeneration and angiogenesis should be further examined. Second, with respect to the signaling mechanisms of bone differentiation induced by SHED-CM, the expression of various bone morphogenetic proteins was confirmed in our previous investigation of cytokine assays. Our data strongly suggest that SHED-CM may enhance osteogenesis, by confirming that the gene expression levels of *ALP*, *Runx2*, and *OCN* and the levels of ALP protein affect the metabolism of bone differentiation. However, it has been difficult to elucidate the detailed signaling mechanisms of bone differentiation. This study suggests that this might be due to not only the participation of simple cytokines but also the involvement of mRNA and matrix proteins, which may induce complex signaling mechanisms during osseous differentiation. 

In recent years, the presence of exosomes contained in the culture supernatant has attracted attention. Exosomes are vesicles composed of lipid double membranes secreted by cells that migrate between cells and transport various molecules [45,46]. Thus, exosomes can be extracted from the cell culture supernatant and serum [47]. Regenerative effects of exosomes have been reported in the bone, muscle, nerve, liver, lung, and periodontal tissues [48,49,50,51,52,53,54]. In addition, exosomes from BMSCs have been reported to enhance osteoblast differentiation and the gene expression of osteogenic markers and promote osteogenesis in rats with skull bone defects [55]. Considering the isolation method for SHED-CM used in this study, it is difficult to independently investigate the effects of exosomes. Hence, to elucidate the effect of SHED-CM on bone regeneration, it is essential to conduct further studies to determine the factors contained in SHED-CM, such as exosomes, microRNAs, and cytokines, and elucidate the factors that promote bone regeneration.

In conclusion, this study revealed that VEGF-A plays an important role in promoting angiogenesis mediated by SHED-CM. Furthermore, SHED-CM promoted the growth of hBMSCs and MC3T3-E1 cells and upregulated the expression of angiogenic and bone differentiation markers in these cells. Thus, SHED-CM efficiently promoted bone regeneration and angiogenesis, without the aid of other biological components. Given the invasion-free approach of the isolation of SHEDs, easy isolation of SHED-CM, and potential treatment strategy via the injection of SHED-CM, SHED-CM may serve as a valuable tool in bone regeneration therapy.

## Figures and Tables

**Figure 1 cells-11-03222-f001:**
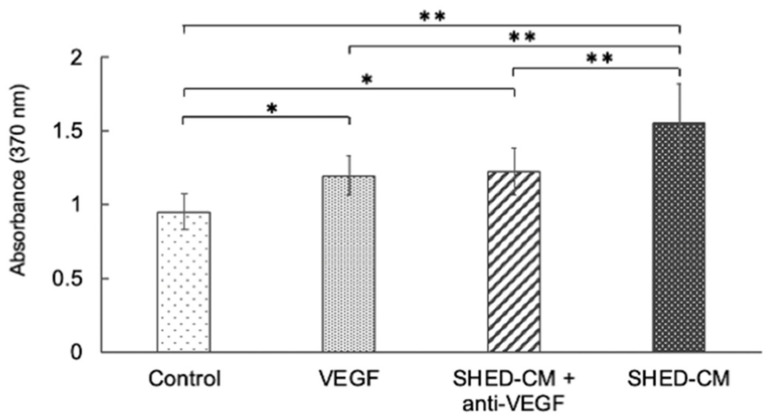
Effect of VEGF-A in SHED-CM on the proliferative ability of HUVECs. The cells in the SHED-CM group exhibited significantly higher proliferative ability than those in the SHED-CM + anti-VEGF-A neutralizing antibody-supplemented (hereafter termed SHED-CM + anti-VEGF-A), VEGF-A, and control groups. Cells in the SHED-CM + anti-VEGF-A and VEGF-A groups showed significantly higher proliferative ability than those in the control group. (*n* = 12, ** *p* < 0.01, * *p* < 0.05). SHED-CM, human deciduous dental pulp-derived mesenchymal stem cell-derived culture supernatant; VEGF-A, vascular endothelial growth factor-A; HUVECs, human umbilical vein endothelial cells.

**Figure 2 cells-11-03222-f002:**
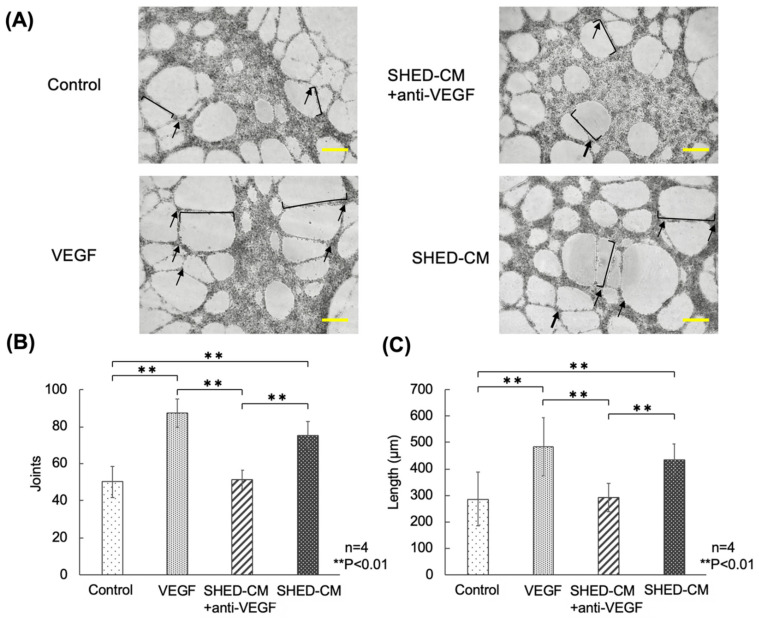
Effects of VEGF-A in SHED-CM on HUVEC lumen assembly. (**A**) Luminal view of HUVECs in the four groups. Lumen formation was observed more frequently in the SHED-CM- and VEGF-A-supplemented groups. Scale bar = 300 μm. (**B**) Vascular bifurcation in the four groups. (**C**) Vessel length in the four groups (*n* = 4, ** *p* < 0.01). Black lines indicate vessel length; black arrows indicate joints. SHED-CM, human deciduous dental pulp-derived mesenchymal stem cell-derived culture supernatant; VEGF-A, vascular endothelial growth factor-A; HUVECs, human umbilical vein endothelial cells.

**Figure 3 cells-11-03222-f003:**
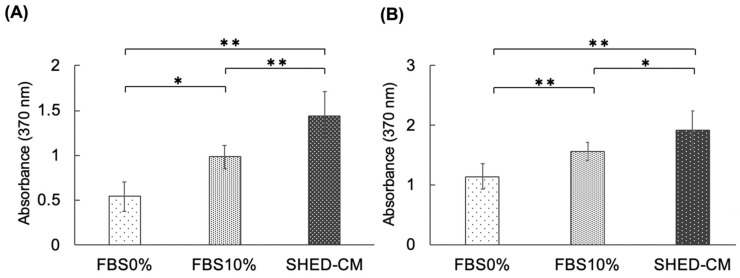
Effect of SHED-CM on the proliferative ability of hBMSCs and MC3T3-E1 cells. (**A**) In hBMSCs, the SHED-CM group presented significantly higher proliferative ability than the serum-free (0% FBS) and 10% FBS groups. A significantly higher cell proliferative ability was observed in the 10% FBS group than in the 0% FBS group. (**B**) In MC3T3-E1 cells, the SHED-CM group showed significantly higher cell proliferative ability than the 0% and 10% FBS groups. Significantly higher cell proliferative ability was observed in the 10% FBS group than in the 0% FBS group. *n* = 12, ** *p* < 0.01, * *p* < 0.05. hBMSCs, human-bone-marrow mesenchymal stem cells; SHED-CM, human deciduous dental pulp-derived mesenchymal stem cell-derived culture supernatant; FBS, fetal bovine serum.

**Figure 4 cells-11-03222-f004:**
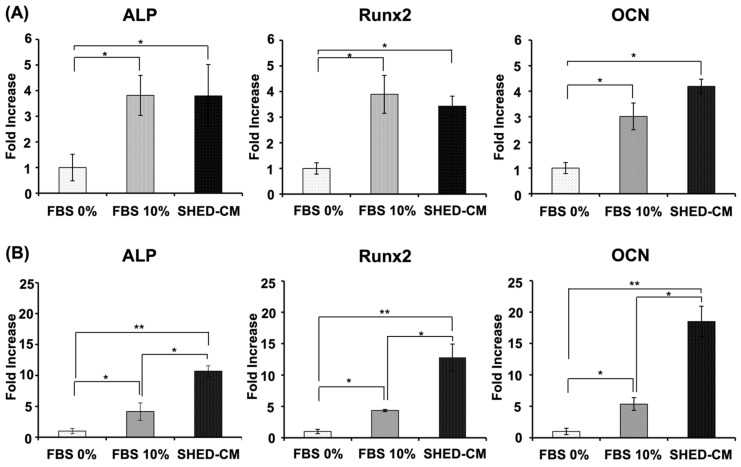
Effects of SHED-CM on hBMSCs and MC3T3-E1 gene expression. The expression of *ALP*, *Runx2,* and *OCN* was significantly enhanced in (**A**) hBMSCs and (**B**) MC3T3-E1 cells supplemented with SHED-CM or 10% FBS, compared with the 0% FBS group. The expression of *ALP*, *RUNX2*, and *OCN* was significantly enhanced by SHED-CM supplementation in MC3T3-E1 cells compared with that in the 10% FBS groups. (*n* = 3, ** *p* < 0.01, * *p* < 0.05). SHED-CM, human deciduous dental pulp-derived mesenchymal stem cell-derived culture supernatant.

**Figure 5 cells-11-03222-f005:**
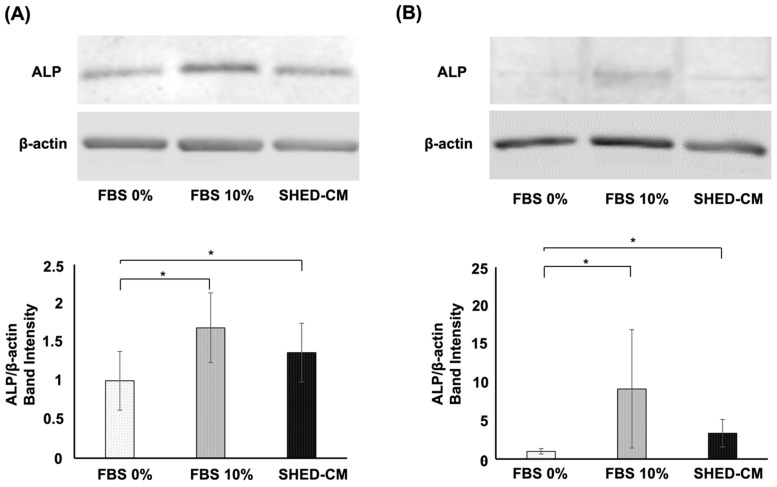
Effects of SHED-CM on ALP protein expression in hBMSC and MC3T3-E1 cells. ALP protein levels were enhanced by exposure to SHED-CM or 10% FBS in (**A**) hBMSCs and (**B**) MC3T3-E1 cells, compared with the 0% FBS group. β-actin expression was used as a loading control. (*n* = 3; * *p* < 0.05) SHED-CM, human deciduous dental pulp-derived mesenchymal stem cell-derived culture supernatant.

**Table 1 cells-11-03222-t001:** Details of the primers used in this study.

Gene		Sequence ( 5′ → 3′)
GAPDH	Forward	CCA CTC CTC CAC CTT TGA
	Reverse	CAC CAC CCT GTT GCT GTA
Runx2	Forward	CCA GAT GGG ACT GTG GTT TAC TG
	Reverse	TTC CGG AGC TCA GCA GAA TAA
ALP	Forward	ATG GTG GAC TGC TCA CAA C
	Reverse	GAC GTA GTT CTG CTC GTG GA
OCN	Forward	GCA GAG TCC AGG AAA GGG TG
	Reverse	GTC AGC AAC TCG TCA CAG

## Data Availability

Not applicable.

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
