# Peer review of "Effects of Human Deciduous Dental Pulp-Derived Mesenchymal Stem Cell-Derived Conditioned Medium on the Metabolism of HUVECs, Osteoblasts, and BMSCs"

_cells, 2022, doi:10.3390/cells11203222_

Round 1
Reviewer 1 Report
Manuscript ID: cells-1836486
Remarks to the Author:
In their manuscript entitled "Effects of human deciduous dental pulp-derived mesenchymal stem cell-derived conditioned medium on metabolism of HUVECs, osteoblasts, and BMSCs," Kunimatsu et al reported that SHED-CM induced higher proliferation in HUVEC cells promoting angiogenesis vie VEGF-A. They showed that SHED-CM thereby promoted the growth of human bone marrow mesenchymal stem cells (hBMSCs) and mouse calvarial osteoblastic cells (MC3T3-E1 cells). Furthermore, they found that SHED-CM upregulated the expression of bone alkaline phosphatase (ALP), osteocalcin (OCN) and Runt-related transcription factor 2 (RUNX2) in hBMSCs and MC3T3-E1 cells. In this work, the authors concluded that SHED-CM efficiently promote angiogenesis resulting in bone regeneration.
Potentially, this is an important study revealing a novel function of SHED-CM for bone regeneration. However, I have the following concerns about the experimental procedures and the quality of the figures that need to be addressed to make the paper suitable for publication:
1. The RT-PCR experiment lacks the negative controls, i. e. omission of reverse transcription to show that the PCR products are not derived from DNA contamination.
2. The demonstration of the mRNA-expression of ALP, OCN and RUNX2 in hBMSCs and MC3T3-E1 cells is not sufficient. Proof of ALP, OCN and RUNX2 presence should be obtained at the protein levels. The authors should use the hBMSCs and MC3T3-E1 cell cultures to do Western blot of ALP, OCN, RUNX2 and VEGF, if this is feasible.
3. The graphics and the labeling are not sharp. The illustrations should be created in higher resolutions. Especially, the quality of Fig. 2A should be improved.
4. In Figures 1 and 2, "VEGF" was written in the legend as VEGF-A, but in the figure as VEGF. In all images, VEGF should also be written as VEGF-A. In Figures 2, 3, and 4, the images were written with capital letters (A, B) and in the legend with small letters (a, b). In labeling figures and legends with capital and small letters, the authors should remain identical.
Author Response
Response
Reviewer #1
In their manuscript entitled "Effects of human deciduous dental pulp-derived mesenchymal stem cell-derived conditioned medium on metabolism of HUVECs, osteoblasts, and BMSCs," Kunimatsu et al reported that SHED-CM induced higher proliferation in HUVEC cells promoting angiogenesis vie VEGF-A. They showed that SHED-CM thereby promoted the growth of human bone marrow mesenchymal stem cells (hBMSCs) and mouse calvarial osteoblastic cells (MC3T3-E1 cells). Furthermore, they found that SHED-CM upregulated the expression of bone alkaline phosphatase (ALP), osteocalcin (OCN) and Runt-related transcription factor 2 (RUNX2) in hBMSCs and MC3T3-E1 cells. In this work, the authors concluded that SHED-CM efficiently promote angiogenesis resulting in bone regeneration.
Potentially, this is an important study revealing a novel function of SHED-CM for bone regeneration. However, I have the following concerns about the experimental procedures and the quality of the figures that need to be addressed to make the paper suitable for publication:
- The RT-PCR experiment lacks the negative controls, i. e. omission of reverse transcription to show that the PCR products are not derived from DNA contamination.
[Response]: We are grateful for your suggestions to improve our manuscript, and we appreciate your interest in our research. An additional PCR experiment including the negative control was performed as indicated. Newly added information can be found on Page 7-8, Line 275-292 , and Fig.4. We have also added and explained the methods and techniques of PCR to the M&M section. Please check Page 4, Line 171-177.
- The demonstration of the mRNA-expression of ALP, OCN and RUNX2 in hBMSCs and MC3T3-E1 cells is not sufficient. Proof of ALP, OCN and RUNX2 presence should be obtained at the protein levels. The authors should use the hBMSCs and MC3T3-E1 cell cultures to do Western blot of ALP, OCN, RUNX2 and VEGF, if this is feasible.
[Response]: We thank you for providing critical comments and useful suggestions that have helped us improve our manuscript. According to your suggestions, we have carried out the additional western blot analysis experiments for the detection of the protein expression of ALP. This is actually our technical challenge, we already tried since the beginning of the additional experiments to order antibodies for RUNX2 and OCN, unfortunately due to significant delay it is not available until now so we only present the result of Western blotting findings of ALP.
Please check. Revised text: Page 8 and 9, Line 293 - 305.
- The graphics and the labeling are not sharp. The illustrations should be created in higher resolutions. Especially, the quality of Fig. 2A should be improved.
[Response]: Thank you for your suggestion. According to your instructions, Figure 2A was improved with higher resolution and more sharpness.
- In Figures 1 and 2, "VEGF" was written in the legend as VEGF-A, but in the figure as VEGF. In all images, VEGF should also be written as VEGF-A. In Figures 2, 3, and 4, the images were written with capital letters (A, B) and in the legend with small letters (a, b). In labeling figures and legends with capital and small letters, the authors should remain identical.
[Response]: Thank you for your valuable advice. Edits to figures 1 and 2 legends were carried out accordingly. Further, Figures 2, 3, and 4 letters were unified (capitalized) in both the figures and legends. Please see Page 6, 7 and 8 Line 249-252, 266-278 and 286-291.
Reviewer 2 Report
In the manuscript entitled “Effects of human deciduous dental pulp-derived mesenchymal stem cell-derived conditioned medium on metabolism of HUVECs, osteoblasts, and BMSCs”. The authors assessed the effects of human SHED-CM on the properties of various cell types showed that SHED-CM promoted the metabolism of HUVECs, MC3T3-E1 cells, and hBMSCs.
Comments:
1. What is the significance of using osteogenic cell line with HUVEC?
2. Authors did not show any molecular mechanisms of the effect of SHED-CM on metabolism of HUVECs, osteoblasts, and BMSCs.
3. Various research articles have been published regarding the Effects of SHED- conditioned medium on HUVECs, osteoblasts, and BMSCs as listed below. What is the strong point of this article? Just examples:
· SPHM de Cara2019: Angiogenic properties of dental pulp stem cells conditioned medium on endothelial cells in vitro and in rodent orthotopic dental pulp regeneration.
· M Kato · 2020: Secreted Factors from Stem Cells of Human Exfoliated Deciduous Teeth Directly Activate Endothelial Cells to Promote All Processes of Angiogenesis.
· HT Vu · 2022: Investigating the Effects of Conditioned Media from Stem Cells of Human Exfoliated Deciduous Teeth on Dental Pulp Stem Cells
4. Line 17: “in them” is not appropriate word.
5. Line 84: Authors need to briefly state the characterization criteria as stem cells and the findings for SHED.
6. Line 135-136: The sentence needs to be rewrite (SHED-CM group, 10% FBS group, and 0% FBS group).
7. How did authors confirm that there were no genomic DNA contaminations in the isolated total RNA?
8. In effects of SHED-CM on hBMSC and MC3T3-E1 Gene Expression, negative control needs to be involved in the experiment.
9. English editing and proofreading through the entire manuscript need to be done.
Author Response
Reviewer #2
Comments and Suggestions for Authors
In the manuscript entitled “Effects of human deciduous dental pulp-derived mesenchymal stem cell-derived conditioned medium on metabolism of HUVECs, osteoblasts, and BMSCs”. The authors assessed the effects of human SHED-CM on the properties of various cell types showed that SHED-CM promoted the metabolism of HUVECs, MC3T3-E1 cells, and hBMSCs.
Comments:
- What is the significance of using osteogenic cell line with HUVEC?
[Response]: We are very grateful for your suggestions that have helped us improve our manuscript, and we appreciate your interest in our research.
There have long been advocates of methods for organizational regeneration that combine the three elements of the ‘Tissue Engineering Triad’, namely cells, scaffold, and growth factors (Langer R & Vacanti, J. P. (1993). Tissue engineering. Science). Recently, however, some researchers have added oxygen and [blood vessels] that provide nutrition to form the ‘Tissue Engineering Quad’. Our previous study demonstrated that when SHED-CM and SHED grafting was performed to treat bone defects in the parietal bone of immuno-compromised mice, accelerated mature osteogenesis was observed in the SHED-CM group compared with that in the SHED and control groups, accompanied by high expression of VEGF and CD31 and enhanced angiogenesis (Hiraki et al., Oral Dis. 2020). In this regard, it is assumed that SHED-CM may affect both angiogenesis and osteogenic bone metabolism. To better define the background on which SHED-CM have led to the investigation of angiogenesis and bone turnover, we have added this content to the Introduction section. Please check Page 2, Line 80-84.
- Authors did not show any molecular mechanisms of the effect of SHED-CM on metabolism of HUVECs, osteoblasts, and BMSCs.
[Response]: Thank you for your suggestion, In this study, it has been revealed that VEGF-A blockade in SHED-CM suppressed the proliferative ability and angiogenic potential of HUVECs, indicating that VEGF in SHED-CM contributes to angiogenesis. Regarding the effect of SHED-CM on bone differentiation, we determined, via ALP western blot analysis (according to Reviewer #1), that SHED-CM enhances the protein expression of ALP (main osteogenic markers). Regarding the signaling mechanisms of bone differentiation in SHED-CM, the expression of various bone morphogenetic proteins was confirmed in the examination of cytokine assays we previously investigated (Hiraki et al., Oral Dis. 2020). It is strongly suggested that SHED-CM may enhance osteogenesis by confirming that gene-expression and protein-expression of ALP affect the metabolism of bone differentiation. However, it has been difficult to elucidate the detailed signaling mechanisms of bone differentiation. This study suggests that this may be because they involve not only the participation of simple cytokines but also that of mRNA and matrix bezicle, which may have complex signaling mechanisms for osseous differentiation. We have added this research limitation to the Discussion section. Please check Page 10, Line 364-378 .
- Various research articles have been published regarding the Effects of SHED- conditioned medium on HUVECs, osteoblasts, and BMSCs as listed below. What is the strong point of this article? Just examples:
- SPHM de Cara · 2019: Angiogenic properties of dental pulp stem cells conditioned medium on endothelial cells in vitro and in rodent orthotopic dental pulp regeneration.
- M Kato · 2020: Secreted Factors from Stem Cells of Human Exfoliated Deciduous Teeth Directly Activate Endothelial Cells to Promote All Processes of Angiogenesis.
- HT Vu · 2022: Investigating the Effects of Conditioned Media from Stem Cells of Human Exfoliated Deciduous Teeth on Dental Pulp Stem Cells
[Response]: I would like to thank you for providing us with information on other papers.
We believe that the three main strengths of this research report over articles informed by the reviewers are as follows: To begin with, in order to elucidate the angiogenic potential of SHED-CM, we used HUVEC and evaluated the bone metabolic potential by using MT3T3 E1 cells and hBMSCs cells with three cell lines in total. Also, We believe that there is a novelty in our validation using a SHED-CM that is enriched 20-fold.
In addition, we believe that SHED-CM's ability to metabolize bone has advantages over other papers not only in terms of gene expression but also in terms of specific protein expression using western blot analysis.
The papers for which the information was provided were added to the Discussion section as reference articles. Please see Page 10, Line 333-336,365-369.
- Line 17: “in them” is not appropriate word.
[Response]: Thank you for your advice. We have deleted “in them” and revised the sentence accordingly. Please check.
- Line 84: Authors need to briefly state the characterization criteria as stem cells and the findings for SHED.
[Response]: According to the International Society for Cellular Therapy, the criteria for defining human MSCs are as follows: (1) Cells adhere to the plastic container under standard culture conditions; (2) The cells are positive for CD73, CD90, and CD105 and negative for CD11b or CD14, CD19 or CD79α, CD34, CD45, and HLA-DR; and (3) cells have the ability to differentiate into osteoblasts, cartilage cells, and fat cells.
(1) We confirmed that the cells isolated from the dental pulp were adherent when cultured in a dish (CLS430165 Corning tissue-culture treated culture dishes).
(2) In a previous study (Nakajima et al., 2018), in which cells were isolated from the pulp in exactly the same manner as that in this study, the cells were positive for CD29, CD44, CD73, CD105, CD146, and STRO-1 and negative for CD34 and CD271. Furthermore, flow cytometry (Rikitake et al., 2021), as a preliminary experiment, confirmed that the cells were positive for CD146, CD44, CD90, CD29, and CD73 and negative for CD34.
(3) Our previous study [14] also demonstrated that the isolated SHEDs have the ability to differentiate into osteoblasts, cartilage cells, and fat cells. In this study, the cells were isolated and cultured in exactly the same manner. Moreover, the cells from Rikitake et al [25] were used and cultured in the exact same manner; therefore, they were defined as SHEDs.
Additional information is provided in the Materials & Methods section. Please check Page 2 and 3, Line 95-109 .
- Line 135-136: The sentence needs to be rewrite (SHED-CM group, 10% FBS group, and 0% FBS group).
[Response]: Thank you for your suggestion. This section was rephrased as indicated. Please see Page 4, Line 158.
- How did authors confirm that there were no genomic DNA contaminations in the isolated total RNA?
[Response]: We are grateful for your question. When used for PCR, it is recommended that A260/A280 ratio be between 1.5 and 2.0.
Gryson 2010a. Effect of food processing on plant DNA degradation and PCR-based GMO analysis: a review. Anal Bioanal Chem 396, 2003–2022
This study was performed using samples with an A260/A280 ratio of 1.5–2.0. In order to clearly explain DNA contaminations in the isolated total RNA methods, we have added and explained the methods and techniques of PCR to the Materials & Methods section. Please check Page 4, Line 171-177 .
- In effects of SHED-CM on hBMSC and MC3T3-E1 Gene Expression, negative control needs to be involved in the experiment.
[Response]: We are grateful for your suggestions to improve our manuscript, and we appreciate your interest in our research. An additional PCR experiment including the negative control was performed as indicated. Newly added information can be found on Page 7-8, Line 275-292 , and Fig.4
- English editing and proofreading through the entire manuscript need to be done.
[Response]: Thank you for your insightful suggestion. The manuscript has been proofread by a native English speaker thanks to Editage editing services.
Reviewer 3 Report
The authors reported that VEGF in SHED-CM affected luminal architecture, proliferative ability, and angiogenic potential of HUVEC cells. Also SHED-CM affected the proliferation of hBMSCs and MC3T3-E1 cells as well as ALP, OCN and RUNX2 in these cells. From these findings, the authors claimed the potential benefits of SHED-CM in bone tissue regeneration.
The findings regarding paracrine action of cytokines were reported in conditioned medium (CM) in MSC-CM in 2004, 2007, 2008, 2011, 2012, 2013, 2014, 2018 etc as well as in SHED-CM.
This reviewer has the following concerns.
Figure 1. Is SHED-CM 20x concentrated in Figure 1? The concentration of VEGF in SHED-CM should be measured.
Figure 2. Is SHED-CM 20x concentrated in Figure 2?The concentration of VEGF in SHED-CM should be measured. The authors need to label the difference in lumen formation such as joints and vessel length among different groups in Figure 2a. Does Figure 2b show the joints? Does Figure 2c showed the vessel length? The authors need to label the joints and vessel length.
Figure 3. The authors need to include the effect of VEGF (10 ng/ml) and SHED-CM+anti-VEGF antibody in Figure 3a and 3b.
Figure 4. The authors need to include the effect of VEGF (10 ng/ml) and SHED-CM+anti-VEGF antibody in Figure 4a and 4b.
The availability of VEGF concentrations in 20x concentrated CM and the data to show a dose response effects on cells might help to verify the functions in lumen formation, cell proliferation and specific bone marker expression.
Author Response
Reviewer #3
Comments and Suggestions for Authors
The authors reported that VEGF in SHED-CM affected luminal architecture, proliferative ability, and angiogenic potential of HUVEC cells. Also SHED-CM affected the proliferation of hBMSCs and MC3T3-E1 cells as well as ALP, OCN and RUNX2 in these cells. From these findings, the authors claimed the potential benefits of SHED-CM in bone tissue regeneration.
The findings regarding paracrine action of cytokines were reported in conditioned medium (CM) in MSC-CM in 2004, 2007, 2008, 2011, 2012, 2013, 2014, 2018 etc as well as in SHED-CM.
This reviewer has the following concerns.
Figure 1. Is SHED-CM 20x concentrated in Figure 1? The concentration of VEGF in SHED-CM should be measured.
[Response]: We are grateful for your suggestions to improve our manuscript, and we appreciate your interest in our research. For Fig. 1, SHED-CM results from 20x concentrated samples.
In our previous study, a multiple cytokine analysis of the protein level in SHED-CM was performed on purified SHED-CM, and the VEGF-A protein level was found to be 161 pg/mL (Hiraki et al., Oral Dis. 2020). Therefore, in this study, the supernatant was prepared in exactly the same way as per the previous experiments, resulting in a final density of 3.2 ng/mL.This is further described in the Discussion section. Please check Page 10, Line 332-336.
Figure 2. Is SHED-CM 20x concentrated in Figure 2?The concentration of VEGF in SHED-CM should be measured. The authors need to label the difference in lumen formation such as joints and vessel length among different groups in Figure 2a. Does Figure 2b show the joints? Does Figure 2c showed the vessel length? The authors need to label the joints and vessel length.
[Response]: Thank you for your valuable suggestions.
Samples of SHED-CM in Fig. 2 are from 20-fold enrichment.
As described above, the concentration of SHED-CM containing VEGF-A protein is 3.2 ng/mL.
In accordance with the reviewer's indications, for Figures 2a and b, the lengths of the joints and vessels were filled out in the figure and modified for clarity. Please see Page 6, Line248-249, Page 7 line 252.
Figure 3. The authors need to include the effect of VEGF (10 ng/ml) and SHED-CM+anti-VEGF antibody in Figure 3a and 3b.
Figure 4. The authors need to include the effect of VEGF (10 ng/ml) and SHED-CM+anti-VEGF antibody in Figure 4a and 4b.
[Response]: Thank you for your valuable suggestions. For the efficacy of anti-VEGF antibody, 10ng/mL of anti-VEGF-A antibody were used to block VEGF-A in accordance with previous in vitro reports.
In this study, since the participation of VEGF-A in angiogenesis is strongly suggested for cells of the vascular system, we conducted VEGF-A blocking and conducted studies. This has revealed that VEGF-A blockade in SHED-CM suppressed the proliferative ability and angiogenic potential of HUVECs, indicating that VEGF in SHED-CM contributes to angiogenesis.
Regarding the effect of SHED-CM on bone differentiation, we determined that SHED-CM enhances the protein expression of ALP main osteogenic markers by performing western blot analysis and additional PCR experiments including the 0% FBS group, according to the suggestion of Reviewer #1.
The reason why VEGF could not be blocked against bone-related cells was that it was difficult to determine which protein expression factors were potent because of the high abundance of proteins related to bone formation.
Regarding the signaling mechanisms of bone differentiation in SHED-CM, the expression of various bone morphogenetic proteins was confirmed in the examination of cytokine assays we previously investigated (Hiraki et al., Oral Dis. 2020). Therefore, we decided to investigate the effect of adding SHED-CM to bone-related cells on bone differentiation. It was strongly suggested that SHED-CM may enhance osteogenesis by confirming the gene-expression of ALP, Runx2, and OCN and protein-expression of ALP to affect the metabolism of bone differentiation. However, it has been difficult to elucidate the detailed signaling mechanisms of bone differentiation. The reason is that this study suggests not only the participation of simple cytokines but also the participation of mRNA and matrix bezicle, which may have complex signaling mechanisms of osseous differentiation. We have added this research limitation to the Discussion section. Please check Page 10 , Line364-368.
The availability of VEGF concentrations in 20x concentrated CM and the data to show a dose response effects on cells might help to verify the functions in lumen formation, cell proliferation and specific bone marker expression.
[Response]: I would like to thank you for your valuable suggestions that helped improve this study.
In the present study, based on previous reports, SHED-CM was enriched 20-fold for the in vitro examination. However, the dose–response efficacy for SHED-CM enrichment has not been investigated in depth. In addition, safety has not been confirmed. In the future, the safety of the concentration of SHED-CM as well as the optimum condition of bone regeneration and angiogenesis should be further examined. We have added this research limitation to the Discussion section. Please check Page 10 , Line364-368.
Round 2
Reviewer 2 Report
Authors addressed most of the comments.
Author Response
Comments and Suggestions for Authors
Authors addressed most of the comments.
[Response]: We are grateful for your suggestions to improve our manuscript.
As per Reviewer 3’s instructions, we have changed the presentation of Figures 2 and 4.
The manuscript has been proofread by a native English speaker, by availing the services of Editage. Modified text is highlighted in green. Please check the revised manuscript accordingly.
Reviewer 3 Report
The authors have improved the manuscript in this submission. However, there are still two issues regarding Fig. 2 and Fig. 4 needed to be clarified.
In Fig. 2A, the authors labeled the numbers of the joints and the lengths of the vessels.
Comment: It is good.
It would be helpful if the authors can provide the enlarged image of vessel bifurcation, joints and vessels.
In Fig. 2, (B) Vascular bifurcation and vessel length in the four groups. (C) Vascular bifurcation and vessel length in the four groups (n=4, ** P<0.01)
Comment: (B) and (C) both have the same statements. Does “(B) Vessel joints in the four groups” better reflect Fig. 2B? Does “(C) Vessel length in the four groups” better reflect Fig. 2C?
Line 273: In hBMSCs, the expression of ALP, Runx2, and OCN was markedly upregulated prior
to the addition of 10% FBS and SHED-CM compared with that in the 0% FBS group (Fig-
ure 4A).
Comment: This statement is confusing. Why did the authors said that the expression of ALP, Runx2 and OCN was upregulated prior to the addition of 10% FBS and SHED-CM compared with that in the 0% FBS griyo? Did Fig. 4A show that the addition of 10% FBS and SHED-CM for 48 hours resulted in upregulation of ALP, Runx2, and OCN?
Line 277: In MC3T3-E1 cells, the expression of ALP, Runx2, and OCN was markedly en-
hanced prior to the addition of SHED-CM and 10% FBS compared with that in the 0% FBS
group (Figure 4B).
Comment: This statement is confusing too. This reviewer has the same concern as that in Fig. 4A.
Author Response
Reviewer 3
Response
Comments and Suggestions for Authors
The authors have improved the manuscript in this submission. However, there are still two issues regarding Fig. 2 and Fig. 4 needed to be clarified.In Fig. 2A, the authors labeled the numbers of the joints and the lengths of the vessels.
Comment: It is good.It would be helpful if the authors can provide the enlarged image of vessel bifurcation, joints and vessels.
[Response]: We are grateful for your suggestions to improve our manuscript. Unfortunately, it was difficult to take a magnified photograph because the corresponding study was completed, and the plates had already been disposed of. We also searched for magnified images of the picture in Figure 2A, but we were unable to find them. Therefore, it is not possible to provide an enlarged image of these photographs this time.
Thank you for your understanding.
In Fig. 2, (B) Vascular bifurcation and vessel length in the four groups. (C) Vascular bifurcation and vessel length in the four groups (n=4, ** P<0.01)
Comment: (B) and (C) both have the same statements. Does “(B) Vessel joints in the four groups” better reflect Fig. 2B? Does “(C) Vessel length in the four groups” better reflect Fig. 2C?
[Response]: Thank you for your valuable suggestions. For Figure 2B, four groups of vascular bifurcations (joint) are shown. For Figure 2C, the results for the length of the vessels in the four groups are shown. Please check the revised manuscript Pages 6, Lines 245–246 .
Line 273: In hBMSCs, the expression of ALP, Runx2, and OCN was markedly upregulated prior to the addition of 10% FBS and SHED-CM compared with that in the 0% FBS group (Figure 4A).
Comment: This statement is confusing. Why did the authors said that the expression of ALP, Runx2 and OCN was upregulated prior to the addition of 10% FBS and SHED-CM compared with that in the 0% FBS griyo? Did Fig. 4A show that the addition of 10% FBS and SHED-CM for 48 hours resulted in upregulation of ALP, Runx2, and OCN?
[Response]: Thank you for your advice. In order to avoid confusion about this issue, this statement was revised as indicated. Please see Page 7, Line 270-272 accordingly.
Line 277: In MC3T3-E1 cells, the expression of ALP, Runx2, and OCN was markedly enhanced prior to the addition of SHED-CM and 10% FBS compared with that in the 0% FBS group (Figure 4B).
Comment: This statement is confusing too. This reviewer has the same concern as that in Fig. 4A.
[Response]: We appreciate your suggestion and apologize for the confusion this might have caused. In order to avoid confusion concerning this issue, the corresponding statement was revised as indicated. Please see Page 7, Line 274-275.
Other modifications
The manuscript has been proofread by a native English speaker, by availing the services of Editage. Modified text is highlighted in green. Please check the revised manuscript accordingly.